# Variance Dichotomy in Feature Spaces of Facial Recognition Systems is a Weak Defense against Simple Weight Manipulation Attacks

**Matthew Bowditch**                                              *Matthew.Bowditch@warwick.ac.uk*
*Mathematics Institute*
*University of Warwick*
*UK*

**Mike Paterson**                                                *M.S.Paterson@warwick.ac.uk*
*Department of Computer Science*
*University of Warwick*
*UK*

**Matthias Englert**                                              *M.Englert@warwick.ac.uk*
*Department of Computer Science*
*University of Warwick*
*UK*

**Ranko Lazić**                                                  *R.S.Lazic@warwick.ac.uk*
*Department of Computer Science*
*University of Warwick*
*UK*

**Reviewed on OpenReview:** *https://openreview.net/forum?id=Q1CfO7flwD*

## Abstract

We show that several leading pretrained facial recognition systems exhibit a variance dichotomy in their feature space. In other words, the feature vectors approximately lie in a lower dimensional linear subspace. We demonstrate that this variance dichotomy degrades the performance of an otherwise powerful scheme for anonymity/unlinkability and confusion attacks on facial recognition system devised by Zehavi et al. (2024), which is based on simple weight manipulations in only the last hidden layer. Lastly, we propose a method for the attacker to overcome this intrinsic defense of these pretrained facial recognition systems.

## 1 Introduction

Within the wide and impactful field of identity verification (see e.g. Schroff et al. (2015); Tang et al. (2017); Han et al. (2019); Labati et al. (2019); Liu et al. (2021); Sepas-Moghaddam & Etemad (2023)), a prominent and successful area is **facial recognition** (see e.g. Taigman et al. (2014); Schroff et al. (2015); Wang et al. (2018); Zhong et al. (2021); Deng et al. (2022)).

Most leading facial recognition systems are based on the **"Siamese" deep neural network architecture** (Bromley et al., 1993), which processes pairs of input images through the same network to produce corresponding vectors in the final feature space. The two feature vectors are then compared by computing the cosine of their angle (or an essentially equivalent similarity measure): if this is larger than a predetermined threshold, then the two input images are classified as *matched*, i.e. as of the same individual; and if this is smaller than the threshold, then they are classified as *mismatched*, i.e. as of different individuals.

This approach therefore constitutes **one-shot open-set recognition** (see e.g. Liu et al. (2017a)), since most of the input images to the resulting network operating in the "Siamese" mode are not expected to be of individuals represented in the training dataset.

We show that a number of leading pretrained networks used for facial recognition have the property that the feature vectors approximately lie in a lower dimensional linear subspace of the entire feature space. We identify one of the main causes of the effect: the weight matrix of the last layer has a number of singular values which are significantly smaller than the rest of the singular values. As a result, any vector that the matrix operates on is mapped, approximately, to a lower dimensional subspace $S$ and the dimension of this subspace $S$ corresponds to the number of "large" singular values.

The term "approximately" is key here because, interestingly, we find that the systems we consider still have a high accuracy when we project the feature vectors onto the subspace orthogonal to $S$, suggesting that this subspace still contains useful semantic information, even though, the angle between two feature vectors in the original space is very close to the angle of those vectors after they are projected onto $S$. Put differently, the subspace orthogonal to $S$ plays almost no role in determining whether two input images are classified as matched or mismatched in the original system. Nevertheless, the subspace orthogonal to $S$ on its own still contains meaningful information.

We then investigate the impact of this phenomenon on a powerful scheme for attacking "Siamese" architecture facial recognition systems that was recently devised by Zehavi et al. (2024). The scheme can be used for two different types of attacks:

- either *anonymity/unlinkability*, where every pair of different images of an individual, chosen by the attacker, is classified as mismatched by the compromised system;

- or *confusion*, where every pair of images of two different individuals, chosen by the attacker, is classified as matched by the compromised system.

In this scheme, an attack involves installing a backdoor by performing simple *weight surgery* exclusively on the last network layer (a process the attacker may disguise as limited tuning). This approach does not require modifying an individual's appearance, perturbing input images, accessing the training dataset, or retraining the network.

The greatest strength of the scheme by Zehavi et al. (2024) is arguably that the attacks can be combined arbitrarily. Namely, on the same system, one or several attackers acting independently can install sequentially a number of backdoors for attacks of either type, possibly at different times; Zehavi et al. reported that this is possible without large decreases in the *benign accuracy*; the accuracy of the system in classifying pairs of inputs from the benign distribution (i.e. when there is no adversary) correctly as either matched or mismatched. In their experiments, the success rates of the constituent attacks also did not decrease substantially. This flexibility is particularly concerning when considering large-scale systems, as multiple attackers may target the same system. Furthermore, an individual attacker, e.g. a state actor looking to anonymize agents operating within another country, may have the objective of anonymizing multiple individuals, making the ability to execute concurrent attacks advantageous.

However, Zehavi et al. limited their investigations to at most 10 combined backdoors and to the FaceNet architecture (Schroff et al., 2015) with 512-dimensional feature space,[1] We extend their analysis by examining how the benign accuracy behaves for several leading facial recognition systems as the number of sequentially installed backdoors increases. We demonstrate that the expected monotonic decline in benign accuracy does not hold. Instead, we uncover a surprising **double descent** in each system:

1. Initially, as the number of backdoors increases, the benign accuracy decreases to a local minimum.

2. Then, counterintuitively, the benign accuracy begins to increase before gradually decreasing again towards a global minimum as more backdoors are added.

---

[1] Since each backdoor in the scheme of Zehavi et al. decreases by one the rank of the last network layer, the theoretically maximum number of sequentially installed backdoors is the dimension of the feature space.

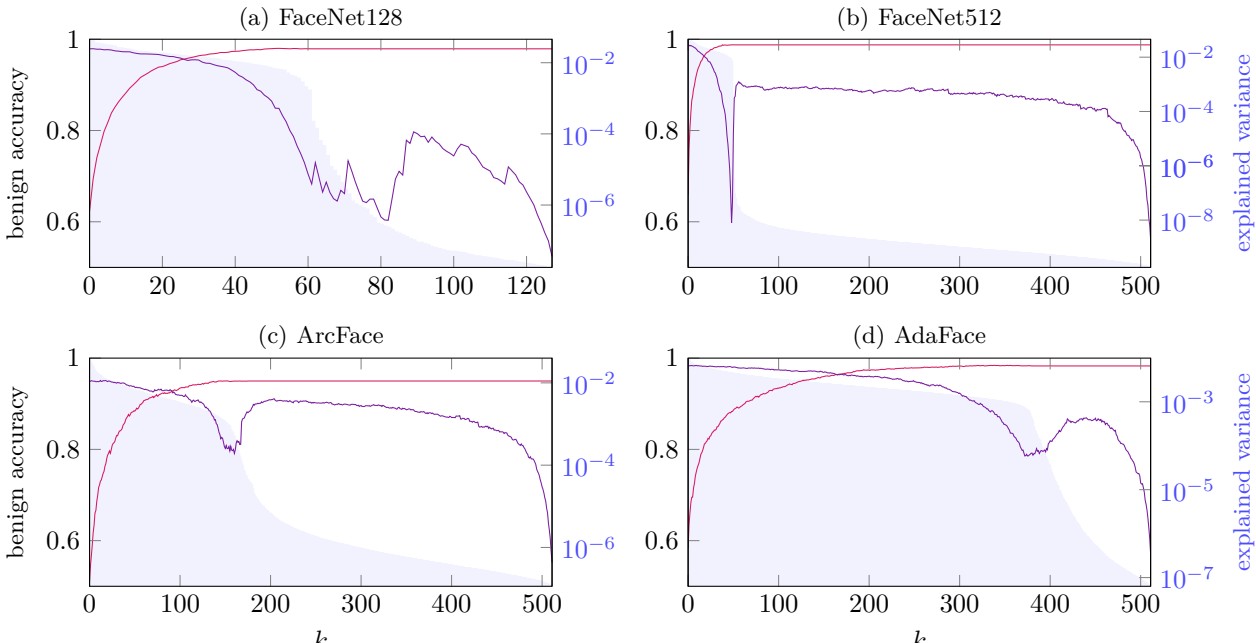

Figure 1: The red curves represent the benign accuracy for each system when the last linear weight matrix $W$ is replaced with $W_k$ (an approximation of $W$ that has rank $k$). The purple curves depict the benign accuracy when $W$ is replaced with $W - W_k$. The shaded blue area shows the explained variance of the principal components of all images from the LFW dataset in feature space, on a logarithmic scale.

We explain this unexpected result by linking it to the aforementioned property of the studied systems: the fact that the feature vectors approximately lie in a lower dimensional subspace. Using this understanding of the feature space, we also provide an improved version of the attack presented in Zehavi et al. (2024).

## 1.1 Our contributions

We perform our analysis on four state-of-the-art pretrained networks: two variants of FaceNet (Schroff et al., 2015) with 512-dimensional and 128-dimensional feature spaces, ArcFace (Deng et al., 2022) and AdaFace (Kim et al., 2022). We do this with the aim to explore vulnerabilities within facial recognition systems, which should be of interest to security researchers and help motivate future defenses. We make the following main contributions:

1. We show that the networks we consider exhibit a **variance dichotomy** in feature space. Namely, principal component analysis of the feature vectors output by the network for inputs from face datasets reveals that the sorted sequence of explained variances has a sharp drop, with the values before the drop being several orders of magnitude larger than those after it. We provide evidence that the dichotomy in explained variances is in large part driven by a corresponding dichotomy in the singular values of the network's final linear weight matrix.

2. We demonstrate that high classification accuracy can be maintained when this final weight matrix is replaced by a low-rank approximation that retains only the largest singular values. Interestingly, when the classifier is instead restricted to operate in the orthogonal complement of this reduced space, using only the directions associated with the discarded, smaller singular values, it still achieves surprisingly high accuracy (see Figure 1). This suggests that while the network primarily relies on a small number of dominant directions for its predictions, the remaining dimensions nonetheless encode meaningful information.

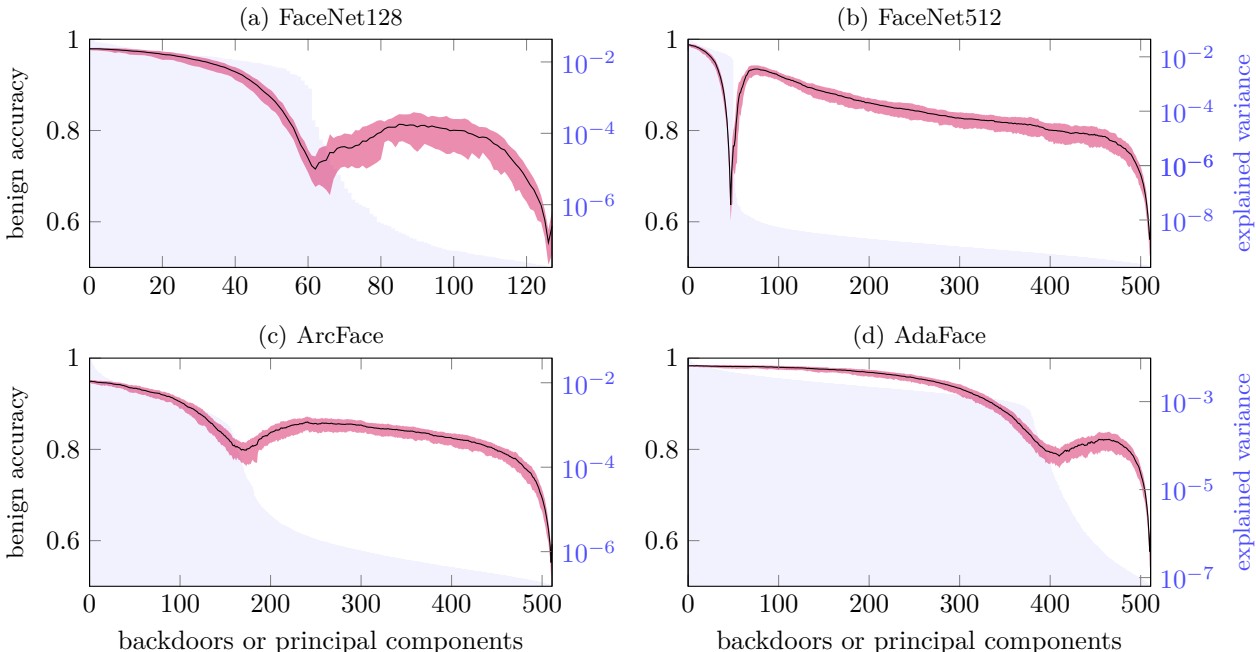

Figure 2: The black curves depict the benign accuracy of the networks as increasing numbers of concurrent Shattered Class backdoors are installed, measured using the LFW dataset. Each backdoor applied is a Shattered Class, with randomly picked individuals from the CelebA dataset (which have not been used in a previously installed backdoor) for which the backdoor is installed. This is repeated ten times and the average is plotted. The red band shows the standard deviation. In addition, the shaded blue area shows the explained variance of the principal components of all images from the LFW dataset in feature space, on a logarithmic scale.

3. We **show false the conjecture**, made implicitly in Zehavi et al. (2024, Section 8), that the benign accuracy decreases as the number of backdoors increases. In the networks we examine, we discover a surprising **double descent** phenomenon. Namely, there are two phases: first, the benign accuracy decreases to a local minimum, which is attained for a number of concurrent backdoors smaller than the dimension of the feature space. Then, as we increase the number of concurrent backdoors towards the maximum number able to be installed, the benign accuracy **increases** before decreasing slowly to a global minimum (see Figure 2). We attribute this phenomenon to the variance dichotomy found in the feature spaces of these networks.

4. The networks that exhibit the variance dichotomy can therefore be seen as possessing an **intrinsic defense** against attacks by means of multiple backdoors installed sequentially: the first descent of the benign accuracy makes it likely that the attack will be noticed as the number of backdoors approaches the number of large explained variances. Using this, we propose a **method for the attacker to overcome** this intrinsic defense.

5. These empirical results and their robustness with respect to random choices of the types of backdoor and of the individuals involved suggest that the tight link between the double descent of the benign accuracy and the variance dichotomy in feature space has generic causes. We analyze in a streamlined setting the counterpart of the benign accuracy, which is defined in terms of cosines of angles between random vectors before and after random orthogonal projections. Its behavior supports the following **explanation of the empirical results**: the feature vectors approximately lie in the subspace spanned by the principal components with the large explained variances, and the first descent of the benign accuracy is governed mainly by their projections onto that subspace; after its dimension

is exceeded by the number of sequentially installed backdoors, it is mainly the projections of the feature vectors onto the principal components with the small explained variances (i.e. the orthogonal complement subspace) which govern the second descent.

## 1.2 Related work

**Intrinsic dimension and simplicity bias.** The variance dichotomy of feature vectors that plays a central role in this paper is conceptually close to notions like the intrinsic dimension of data representations in deep neural networks (Ansuini et al., 2019) and the simplicity bias of their training algorithms (Morwani et al., 2024). However, we are not aware of previous work that focuses on links between such notions and the susceptibility of identity verification systems to multiple anonymity/unlinkability or confusion attacks.

**Weight manipulation attacks.** Closest to the attack scheme of Zehavi et al. (2024) that we investigate in this paper are works on adversarial manipulations of neural network weights, such as Liu et al. (2017b); Dumford & Scheirer (2020); Qi et al. (2022); Bai et al. (2023). With the exception of the single bias attack of Liu et al. (2017b), these approaches rely on an iterative approach, which is not guaranteed to find a good solution; and they require editing network layers other than the last one, which cannot be readily disguised as fine tuning.

**Poisoning attacks.** These attacks typically proceed by targeted poisoning of a dataset on which a network of interest is then trained, and are exemplified by the works of Shafahi et al. (2018); Chen et al. (2019); Lin et al. (2020); Chen et al. (2021); Doan et al. (2021); Sarkar et al. (2022); Xue et al. (2022); Doan et al. (2022); Gao et al. (2023); Jha et al. (2023). Typically, these attacks require corrupting both the data and modifying the corresponding labels although successful backdoor attacks have been implemented by corrupting only the labels (Jha et al., 2023). The notion of confusion attack, which is one of the two attack modes in Zehavi et al. (2024), appears already in Chen et al. (2019). Multiple backdoors on the same model are considered by Lin et al. (2020), however due to the data poisoning approach it seems necessary that the attacker installs them all at the same time and without interference from other attackers. In Doan et al. (2022), a generative trigger function is trained during backdoor injection, enabling the attacker to generate adversarial perturbations for arbitrary input images and target classes.

**Physical adversarial attacks.** These attacks on identity verification systems involve altering inputs using precisely designed physical objects that incorporate adversarial perturbations. Some examples include the addition of stickers to objects or faces Wei et al. (2023), wearing specially designed accessories Sharif et al. (2019); Singh et al. (2021); Zolfi et al. (2023), clothing Hu et al. (2022; 2023); Sun et al. (2023), or makeup Zhu et al. (2019); Yin et al. (2021); An et al. (2023).

**Adversarial defense.** The existence of adversarial examples has led to the development of a number of defense mechanisms aimed at improving the robustness of identity verification systems. Various methods have been proposed, such as including adversarial examples during training (Zheng et al., 2016; Bai et al., 2021), controlling model sensitivity to changes in input (Ross & Doshi-Velez, 2018), or purifying inputs prior to classification (Nie et al., 2022).

## 2 Reducing the dimensionality of feature spaces

In this section, we want to explore the phenomenon that feature vectors produced by a facial recognition system often, approximately, lie in a lower dimensional subspace of the entire feature space. To investigate the properties of feature vectors for a facial recognition network, we first compute a set of feature vectors by applying the network to the images in the LFW dataset (Huang et al., 2008), a public benchmark frequently used for facial recognition systems. We then run a standard principal component analysis (PCA) on this set of feature vectors.[2]

---

[2]Note that we do not adjust the empirical mean of the data to zero before calculating the covariance matrix, since both the cosine similarity metric, which is at the core of the facial recognition systems, and the backdoor installation scheme, which is based on orthogonal projections, work with feature vectors without any mean adjustment.

Table 1: $(\varepsilon, \delta)$-dichotomy values for sets of feature vectors from various combinations of models and datasets.

| Model | LFW | | | | CelebA | | | |
|---|---|---|---|---|---|---|---|---|
| | $\varepsilon$ | $\delta$ | $i$ | $j$ | $\varepsilon$ | $\delta$ | $i$ | $j$ |
| FaceNet128 | 0.102 | 0.502 | 60 | 73 | 0.109 | 0.502 | 57 | 71 |
| FaceNet512 | 0.004 | 0.516 | 48 | 50 | 0.004 | 0.528 | 48 | 50 |
| ArcFace | 0.170 | 0.500 | 131 | 218 | 0.166 | 0.501 | 139 | 224 |
| AdaFace | 0.107 | 0.500 | 377 | 432 | 0.109 | 0.505 | 376 | 432 |

**Definition 1** (Explained variance and principal components). *Suppose we have a set of n feature vectors $\{x_i\}_{i=1}^{n}$ with each $x_i \in \mathbb{R}^d$. Consider the matrix $X$ that contains the $x_i$ vectors as its rows. Let $Q \in \mathbb{R}^{d \times d}$ be the covariance matrix of these vectors, so $Q = X^T X$. Denote the eigenvalues of $Q$ by $\{\lambda_i\}_{i=1}^{d}$ ordered such that $\lambda_1 \geq \lambda_2 \geq \cdots \geq \lambda_d$ and let $\{v_i\}_{i=1}^{d}$, where $v_i \in \mathbb{R}^d$, be the corresponding right eigenvectors. Then we say that the i-th principal component is $v_i$ and that it has explained variance $\hat{\lambda}_i = \lambda_i / \sum_{j=1}^{d} \lambda_j$.*

The principal components form an orthonormal basis and therefore give us a new coordinate system in which we can interpret the feature vectors. Intuitively, the explained variance of the $i$-th principal component then indicates how much variation there is in the data along the $i$-th axis in this the new coordinate system. If $\lambda_i$ is small, this is an indication that the feature vectors, when rewritten in the new coordinate system, tend to have small values in their $i$-th component.

In our experiments, we use the LFW dataset (which contains approximately 13000 images) to compute the set of feature vectors used to find the principal components and explained variances. We do this for the four different facial recognition systems FaceNet128, FaceNet512, ArcFace, and AdaFace. Figure 1 shows the explained variance ordered by magnitude for each principal component as the shaded blue area, using a logarithmic $y$-axis on the right. When considering the sequence of explained variances $\{\hat{\lambda}_i\}_{i=1}^{d}$ we find that for each network, the explained variances have a sharp drop, with the explained variances prior to the drop being several orders of magnitude larger than those following. This suggests that these networks exhibit a variance dichotomy phenomenon, with some explained variances much larger than the others and only few values in between.

We now formalize this variance dichotomy phenomenon using the following definition:

**Definition 2** $((\varepsilon, \delta)$-dichotomy). *Suppose we have a set of d vectors in $\mathbb{R}^d$ and the corresponding set of explained variances $\{\hat{\lambda}_i\}_{i=1}^{d}$. We say this set of vectors exhibits an $(\varepsilon, \delta)$-dichotomy if there exist $i, j \in \{1, \ldots d\}$ with $|i - j| \leq \varepsilon d$ and*

$$\log\left(\frac{\hat{\lambda}_i}{\hat{\lambda}_j}\right) > \delta \log\left(\frac{\hat{\lambda}_1}{\hat{\lambda}_d}\right)$$

Intuitively, we can think of $\delta$ being the relative size of the logarithmic drop in explained variances and $\varepsilon$ being the fraction of the principal components over which this drop occurs. If a set of vectors exhibits $(\varepsilon, \delta)$-dichotomy for a large $\delta$ and small $\varepsilon$, it means there is a significant (multiplicative) drop in explained variance over just a few principal components. To calculate the results in Table 1 we use output feature vectors from each model, found by inputting either the LFW or CelebA datasets. We choose $\delta = 0.5$, as this value guarantees that there can only be a single drop of this magnitude and therefore, we have a true dichotomy. We then find the minimum $\varepsilon$ such that the feature vectors still exhibit a $(\varepsilon, 0.5)$-dichotomy.

The values of $\epsilon$ in Table 1 measure the strength of the dichotomy for each network. For instance, FaceNet512 has $\epsilon = 0.004$, indicating a strong dichotomy. This is because the explained variance exhibits a relative logarithmic drop of 0.5 over just 0.4% of its principal components. This observation aligns with the shaded blue region in Figure 1(b), where a significant drop in explained variance occurs between indices 48 and 50.

**Approximate low dimensionality of feature vectors.** If some explained variances of a set of feature vectors are very small, this means that the vectors, approximately, lie in a linear subspace of the entire feature space. This is because when the vectors are interpreted in the new coordinate system given by the

principal components, the $i$-th component of the feature vectors have variance $\lambda_i$. Hence, if $\lambda_i$ is very small compared to $\sum_{j=1}^{d} \lambda_j$, the projection of the feature vectors onto that component tend to be small, indicating that this direction contributes little to the overall structure of the data.

What causes the feature vectors to approximately lie in a lower-dimensional subspace? The feature vectors result from multiplying the output of the penultimate layer by the final linear weight matrix. An analysis of the singular values of this matrix reveals a distinct dichotomy in their magnitudes, as illustrated in Figure 6. We find that this dichotomy aligns closely with the observed dichotomy in explained variance. This suggests that the structure of the final weight matrix plays a significant role in causing the variance dichotomy in feature space and this is not an effect primarily caused by the distribution of input images or the transformations performed by earlier layers in the network.

**Reducing feature space dimensionality without loss of accuracy,** The final layer weight matrix $W$ has a very good low rank approximation. The low rank approximation can be obtained by (a) computing the singular value decomposition of $W = U\Sigma V^T$, where $U$ is a $d \times d$ matrix, $\Sigma$ is a rectangular $d \times m$ diagonal matrix containing the singular values, and $U$ is a $m \times m$ matrix (where $m$ is the number of neurons in the penultimate layer), and (b) obtaining $\Sigma_k$, which is equal to $\Sigma$, but only keeps the $k$ largest entry and replaces all other entries by 0. If $k$ is chosen such that only very small entries of $\Sigma$ are replaced by 0, we have a good low rank approximation given by $W \approx W_k := U\Sigma_k V^T$.

If we replace the last layer by $W_k$, all feature vectors now strictly lie in a $k$-dimensional linear subspace $S_k$ of the entire feature space. Our question is how this impacts the accuracy of the facial recognition system.

To calculate the accuracy of a facial recognition system we use the LFW dataset. We employ the standard ten-fold cross-validation procedure, using the predefined splits provided on the LFW homepage. In this process, nine sets——each consisting of 300 matched and 300 mismatched input pairs——are used to optimize the threshold for the cosine similarity metric. We choose the threshold that maximises classification accuracy between matched and mismatched pairs. The remaining set is then used to evaluate the system's accuracy with this optimized threshold. This process is repeated across all ten 9:1 splits, and the final accuracy is reported as the mean of these ten iterations, representing the system's accuracy. The results are shown in Figure 1 for different values of $k$. It shows that the accuracy of the system is not impacted for large enough values of $k$. The removed dimensions were effectively not utilized by the system to classify pairs of images as a match or mismatch.

Surprisingly, the removed dimensions do in fact encode useful features, even if they are effectively ignored for classification in the original system. To demonstrate this, we compute the accuracy when the final layer $W$ is replaced by $W - W_k$ instead of $W_k$. The resulting feature vectors now strictly lie in the subspace that is orthogonal to $S_k$. The resulting accuracies are also shown in Figure 1. For FaceNet128 for example, we can see that for $k = 90$, the accuracy of the system using $W_k$ is the same as for $k = 128$. The removed dimensions made no difference in the classification of images pairs. Nevertheless, the system using $W - W_k$ in the final layer still has an accuracy of nearly 80%. This shows that the removed dimensions by themselves do encode some meaningful features.

## 3 Impact on a facial recognition backdoor attack

We explore a concrete implication of the feature vector properties detailed in Section 2. Namely, the impact on the anonymity/unlinkability and confusion attacks by Zehavi et al. (2024). We start with the following question:

> For several leading facial recognition systems, how does the benign accuracy behave as the number of sequentially installed backdoors increases?

**Two backdoor types.** Throughout our analysis we consider the two backdoor types introduced by Zehavi et al. (2024):

- The *Shattered Class (SC)* backdoor aims to achieve anonymity/unlinkability of an individual chosen by the attacker. The attacker first calculates a projection matrix $P_x$, which projects the feature space in direction $x$, an estimation of the direction of the center of a target individual's class in feature space. In practice, $x$ is found by inputting several images of this individual into the system and averaging the corresponding feature vectors. To install the SC backdoor, the existing weights of the last linear layer $W$ are replaced with $P_x W$.

- The *Merged Class (MC)* backdoor aims to achieve confusion of two target individuals. In this case, we replace $W$ with $P_{x_1 - x_2} W$, where $x_1$ and $x_2$ are found by averaging feature vectors corresponding to the two target individuals.

**Benign Accuracy and Attack Success Rate.** We call the *benign accuracy* of a facial recognition system the accuracy of the system on individuals that were not targeted by an installed backdoor. The goal of the attacker is to keep the benign accuracy high when installing a backdoor, as otherwise the attack may be noticed.

Another metric is the *attack success rate.* For a SC backdoor, we are interested in the success rate in term of two images of the target individual to be declared a mismatch. For the MC backdoor, we are interested in the success rate in terms of two images of the respective target individuals to be declared a match.

**Sequentially installed backdoors.** With each additionally installed backdoor, due to the projection involved, the dimension of the subspace in which the feature vectors lie is reduced by one. As the number of installed backdoors increases, this decreases the performance of the system.

In Figure 2 we show the benign accuracy of each system when applying sequential SC backdoors, with a fixed cosine similarity threshold for each split of the LFW dataset calculated prior to backdoor installation. Figure 7 shows the corresponding attack success rates. In the appendix we provide similar experiments using only merged class backdoors, (Figures 9 and 10), and picking either shattered or merged class uniformly at random for each backdoor (Figures 11 and 12).

Alongside the benign accuracies and the attack success rates for each system, Figures 2 and 7 show as the shaded blue area the explained variances ordered by magnitude for each principal component. Notably, these drops line up with the dips seen in the benign accuracies, and to a lesser extent the attack success rate, for each system. In Section 5 we explore further the link between these two curves.

We observe a double descent phenomenon. The reason lies in the properties of the feature vectors detailed in Section 2. The feature space, approximately, consists of two parts. A lower dimensional subspace $S$ and the space orthogonal to $S$. Initially, the angle between two vectors is mostly determined by the angle between the projections of the vectors onto $S$. As we install more backdoors however, we keep reducing the dimension of $S$. This is the first descent. Eventually, feature vectors entirely lie in the space orthogonal to $S$ at which point this orthogonal space starts to determine what the angle between two such feature vectors is. Since the space orthogonal to $S$ does carry semantic information, the benign accuracy increases. Finally, as we install even more backdoors, the feature vectors eventually lie entirely in a very small dimensional subspace that is no longer sufficient to achieve reasonable benign accuracy.

In Figure 3 we show results of further experiments on the benign accuracy of the four systems, now by installing increasing numbers of shattered class backdoors using the ordered principal components as the backdoor directions. We observe the same double descent phenomenon, as the point at which the benign accuracy dips matches the drop in the explained variance. Again, the surprising result here is that after some projections, when feature vectors in each model are only using the principal components with the small explained variances, we still observe moderate benign accuracy, suggesting that these components are useful for the task of one-shot open-set recognition.

### 3.1 Further experimental details.

**Four facial recognition systems.** Our experiments are conducted on four state-of-the-art facial recognition systems, which are diverse with respect to their architectures and training methods.

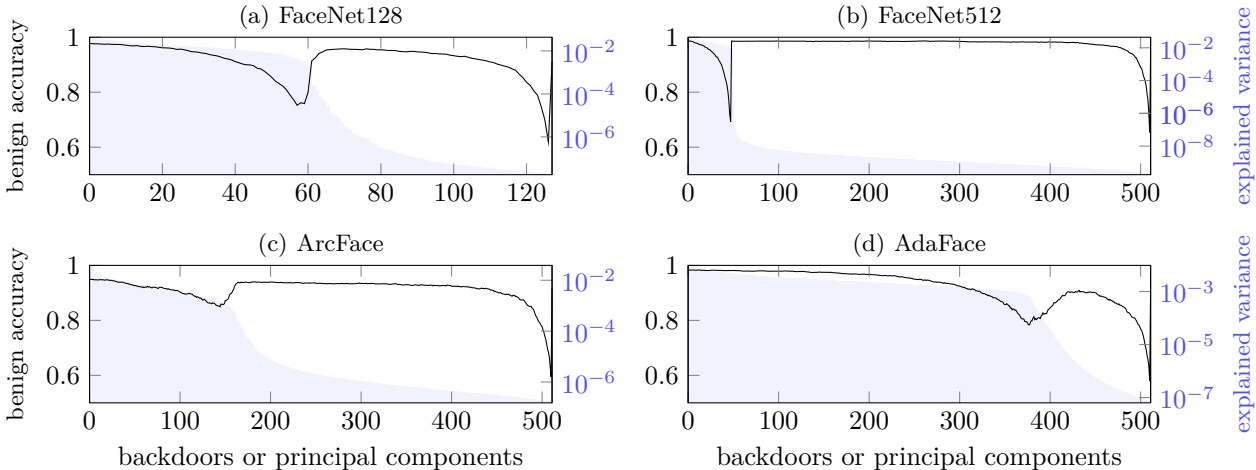

Figure 3: The black curves depict the benign accuracy of the networks as increasing numbers of concurrent "artificial" (i.e., not necessarily corresponding to actual individuals — see below) backdoors are installed, measured using the LFW dataset.Each backdoor applied is a Shattered Class, which uses the principal components, ordered by the magnitude of their principal value, to install the backdoor. In addition, the shaded blue area shows the explained variance of the principal components of all images from the LFW dataset in feature space, on a logarithmic scale.

- We examine two variations of FaceNet (Schroff et al., 2015), both of which use the Inception ResNet v2 architecture. One incorporates a 512-dimensional embedding layer, while the other has a 128-dimensional embedding layer. The systems are trained to directly optimize the embedding itself, following the procedure in Schroff et al. (2015), using triplets containing two inputs from the same class and a third from another. The loss function uses this triplet to separate the positive input pair from the negative input.

- ArcFace (Deng et al., 2022) uses a ResNet-34 architecture, and is trained using the additive angular margin loss function: a hybrid loss function that combines the standard softmax loss with an additive angular margin penalty designed to minimise intra-class distances whilst maximizing inter-class distances.

- AdaFace (Kim et al., 2022) uses a ResNet-100 backbone, and is trained using a loss function that is designed to adapt to a sample's recognizability.

We use the popular facial recognition repository DeepFace[3] for implementations and pretrained weights. This repository wraps several state-of-the-art models and is used throughout our analysis. We align all of our input images prior to input using face and eye detector models provided by OpenCV[4].

**Installing backdoors.** Individuals used for the backdoors are taken from the CelebA dataset (Liu et al., 2015), restricted first to individuals containing over 20 images and restricted further to the 1024 classes with the highest average benign accuracy across the four systems. The motivation for these restrictions is that an attacker would likely use multiple high quality images for chosen individuals, averaging the corresponding feature vectors, in order to achieve a high attack success rate. Note that this dataset differs from the Pins Face Recognition dataset[5] used to install backdoors by Zehavi et al. (2024), since we require a dataset with more individuals so that we can apply many backdoors.

---

[3]https://github.com/serengil/deepface

[4]https://opencv.org/

[5]https://www.kaggle.com/datasets/hereisburak/pins-face-recognition

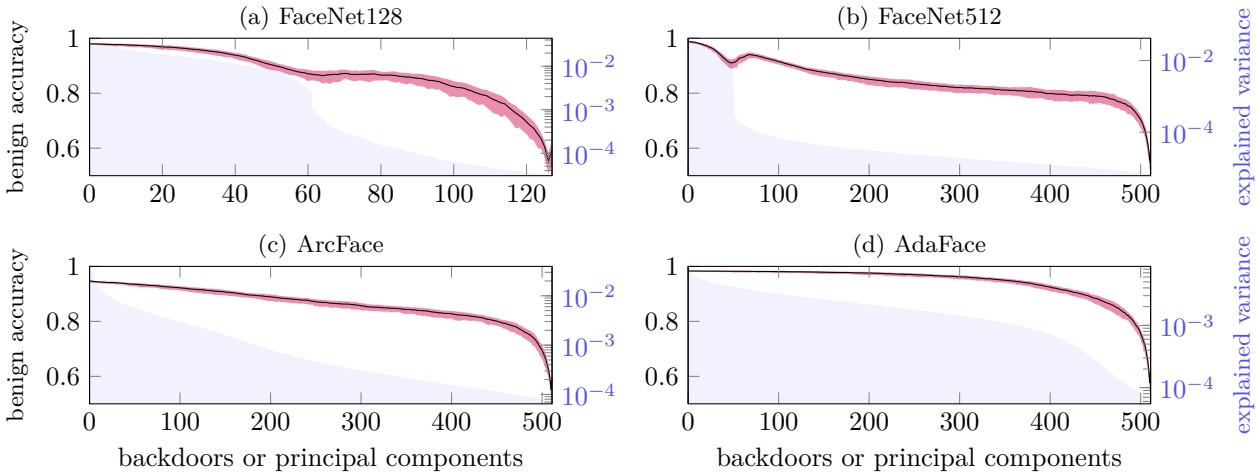

Figure 4: The plots show the results of the same experiments as in Figure 2, but after first performing SVD on the last network layer and replacing the singular values with their mean. As before, the black curve and the red band show the average benign accuracies, and their standard deviation, respectively. The shaded blue area shows the explained variance of the principal components of all images from the LFW dataset, computed after the weight surgery and on a logarithmic scale.

For each system, we run each experiment ten times, each time using a different random order of classes within the restricted CelebA dataset to calculate the backdoor directions. We plot the average result of this ten runs and indicate the standard deviation using a shaded red region around the mean.

**Computing benign accuracy and attack success rate.** To compute the benign accuracy, we employ the standard ten-fold cross-validation procedure detailed in Section 2.

To compute the attack success rate, after each installed backdoor, we average the proportion of pairs of inputs misclassified by the system for each individual backdoor currently installed. We take the mean over each cosine similarity threshold computed when performing the benign accuracy calculation.

A caveat of our methodology is the use of an entire CelebA class to both install the backdoor and calculate the attack success rate. This choice was made due to limitations in the number of images per individual within the CelebA dataset. We found that the results were not significantly different if we split the classes that have moderate to high numbers of images randomly into two halves when installing a backdoor, and then use one half for implementation and the other for subsequent computations of the attack success rate.

## 4 The Revised Method

We identified the dichotomy of the magnitudes of the singular values of the weight matrix of the last layer in the network as the main culprit behind the fast decrease in benign accuracy as more backdoors are installed.

An observant attacker who wishes to implement multiple backdoors may notice a variance dichotomy in the feature space of a facial recognition system, and seek to mitigate this effect to achieve a higher benign accuracy and attack success rate. We show that this is indeed possible, by using a revised method of implementing these backdoors that first applies a transformation to the last linear weight matrix.

To do so, we use a similar method to the backdoor hiding methodology found in Zehavi et al. (2024, Section 7.6), by first performing Singular Value Decomposition (SVD) on the last linear weight matrix and modifying the distribution of the singular values.

Concretely, we:

1. Perform SVD on the last linear weight matrix $W$ to get $W = \sum_{i=1}^{d} \sigma_i u_i v_i$, where the $\sigma_i$'s are the singular values.

2. Find the mean of the singular values, $\sigma = \frac{1}{d} \sum_{i=1}^{d} \sigma_i$.

3. Replace the singular values with $\sigma$ to construct a new weight matrix, $W' = \sum_{i=1}^{d} \sigma u_i v_i$ and replace $W$ with $W'$ in our facial recognition system.

We only perform this operation once, before installing the first backdoor. This reduces the dichotomy of explained variance in feature space, i.e. the feature vectors now belong to a subspace with dimension approximately equal to the full dimension of feature space.

Figures 4 and 8 show experiments using the revised methodology; other details of the experiment remain the same as in Figures 2 and 7. For each LFW split, the threshold is calculated as before using the nine sets. Again, we plot the explained variances for each system with the shaded blue area, calculated after replacing the last linear weight matrix with the modified version. Notably, after performing this modification, the benign accuracy of the unattacked system does not change more than 0.2% in any of the systems we consider. Also, the dips in benign accuracy and attack success rates, present in all four networks in Figures 2 and 7, are now much less steep; in fact, AdaFace and ArcFace see this dip entirely eliminated. Importantly, we observe that this modified attack scheme achieves higher benign accuracies and attack success rates compared with the original for ranges of backdoor numbers around the eliminated dips (see Figure 15 for a clearer comparison of the two methods). This allows the attacker to stealthily and effectively install a significantly greater number of backdoors compared to the previous method.

This modified method does not cause this backdoor to take a significantly longer amount of time to install. The SVD of the last linear weight matrix is quick (up to several seconds in our experiments), since the dimension of the feature space is low.

## 5 Synthetic data

Even with simple synthetic data, we can see how the benign accuracy decreases as the number of backdoors increases and we can obtain plots similar to those in Figures 2 and 4. Each of the four architectures we study has an optimal threshold, $t$, which is compared to the angle between two vectors in feature space to determine whether the two images are classified as belonging to the same individual or not. This threshold is precomputed using the LFW dataset. We use this same threshold to study our synthetic data.

To generate the synthetic data, we compute the cosine of the angles of the vectors in feature space for every pair in the LFW dataset. Using these angles we record the mean, $m$ and $m'$, and standard deviation, $s$ and $s'$, of matching pairs and mismatching pairs, respectively. With this information alone, we generate synthetic data. To obtain a matched pair, we sample $\cos(\varphi)$ of an angle $\varphi$ from a Gaussian distribution (restricted to $[-1, 1]$) with mean $m$ and standard deviation $s$. We then sample two random unit vectors that have an angle of exactly $\varphi$ to one another. To obtain a mismatched pair we do the same but with $m'$ and $s'$.

To simulate the sharp drop in explained variance, we use the above procedure to find a pair of vectors in $k$ dimensions and a pair of vectors in $d - k$ dimensions, where $k$ depends on where the drop in explained variance is. We scale down the second pair by a factor of 1000 and then concatenate both pairs to obtain a pair of vectors in $d$ dimensions. This final pair no longer has an angle of exactly $\varphi$, but the angle is still close to $\varphi$.

We generate 1000 matched and an equal number of mismatched pairs. The benign accuracy on matched (mismatched) pairs is the fraction of pairs of vectors that have an angle of less than (greater than) $t$ to one another. The benign accuracy overall is the average of the benign accuracy on matched and mismatched pairs.

To investigate the effect of installing backdoors, we repeatedly project all vectors in our matched and mismatched pairs in a random direction and recompute the benign accuracy after each projection. The resulting plots for all four architectures can be found in Figure 5. We note that many characteristics from the plots in

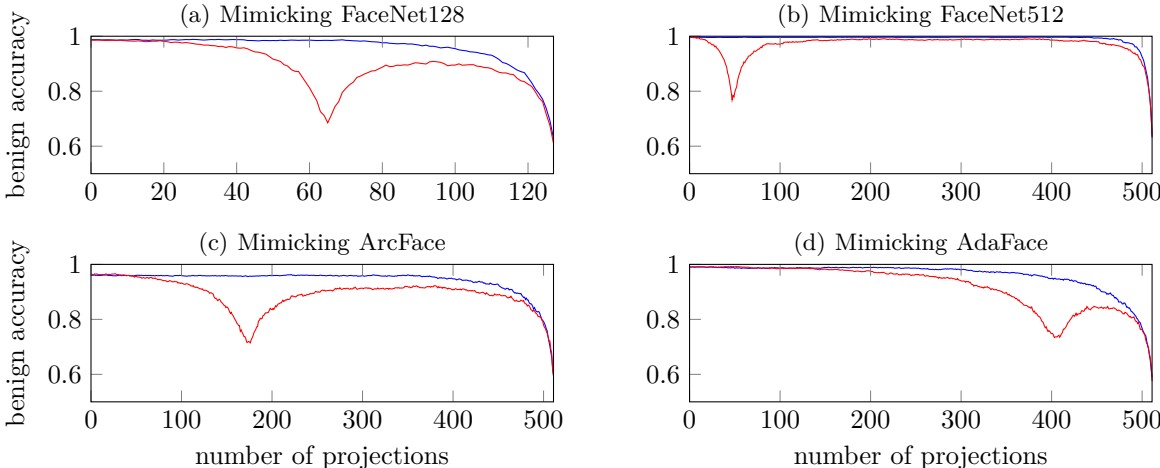

Figure 5: To mimic each of the four networks, we plot the benign accuracy over the synthetic data after repeated orthogonal projections. The direction of projection is chosen in the same way as the vectors in the synthetic data. When simulating the drop in explained variance, we pick a random $k$-dimensional unit vector and concatenate it with a random $(d - k)$-dimensional vector of length $1/1000$. If not simulating the drop, the direction is just a random $d$-dimensional vector. The blue curves show the values for synthetic data without the drop, and they resemble the real benign accuracies when using the modified method, as plotted in Figure 4. The red curves for the first three networks show the values for synthetic data that tries to model the drop in explained variance, and they resemble the real benign accuracies as plotted in Figure 2.

Figures 2 and 4 can already be observed in this simple model involving vectors that are chosen completely at random, except that the distribution of angles between vectors is fixed in a specific way.

## 6 Conclusion

We considered four state-of-the-art deep neural network facial recognition systems and found that all of them exhibit a variance dichotomy in feature space. Specifically, their feature vectors approximately lie in a subspace of significantly lower dimension. We demonstrated that when these classifiers are restricted to the orthogonal complement of this subspace, they still achieve surpisingly high accuracy. This is strongly linked with a double descent phenomenon in the benign accuracy of the networks when installing multiple backdoors - recently devised by Zehavi et al. (2024) for anonymity/unlinkability and confusion attacks. Furthermore, we showed how an attacker can exploit this behavior to significantly increase the number of backdoors that can be installed stealthily and effectively. Supported by an analysis of the impact of random orthogonal projections on angles between random vectors, we also offered an explanation of the link between the double descent of the benign accuracy and the variance dichotomy in feature space.

We envisage that our results will be of interest to researchers in architectures and training of deep neural networks for identity verification, as well as to researchers in neural backdoor/Trojan attacks and defenses.

**Limitations.** Our experiments were performed only on the four pretrained facial recognition systems as detailed in Section 3. However, we expect that the dichotomy property features in other systems and is a common phenomenon. Our study of the link between the dichotomy property and the behavior of the backdoor attacks by Zehavi et al. (2024) is specific to this kind of weight manipulation scheme.

**Future work.** As we did not rely on any aspects of facial recognition other than to work with the standard datasets in this domain, our methodology should be applicable to a broad range of identity verification systems based on the "Siamese" deep neural network architecture. It would be interesting to examine state-

of-the-art systems that work with other biometrics for the variance dichotomy in their feature spaces, the double descent of their benign accuracies under multiple backdoors, and the attack scheme enhancement discussed in Section 4. We envisage that many networks in this space would see comparable results to those shown here, based on the current understanding that the intrinsic dimension of data representations tends to decrease progressively in the final layers of these networks (Ansuini et al., 2019). It would also be interesting to see if other attack schemes are affected by variance dichotomy in feature spaces.

Our explanation of the link between the double descent and the variance dichotomy could be refined and further supported by deriving general theoretical bounds that in particular shed light on the behavior of the benign accuracy around its local minimum.

**Broader impact statement.** This paper presents work whose goal is to advance the field of machine learning. There are many potential societal consequences of our work, none of which we feel must be specifically highlighted here.

Our research on backdoor attacks and defenses of identity verification systems based on deep neural networks is intended to highlight weaknesses, so that those using them better understand the risks involved.

We have shown that relying on noticing reduced accuracy in the system may not be enough to identify a compromised system, even in the event of multiple backdoors. Instead, potential safeguards that avoid this reliance include:

- Automated checking of the rank of the final layer matrix. Each backdoor reduces this by one due to the projection applied.

- Automated checking of variance dichotomy in feature space.

- Using an ensemble of networks to verify identity to mitigate the effect if one is compromised.

### Acknowledgments

This research was supported by the Centre for Discrete Mathematics and Its Applications (DIMAP) at the University of Warwick and the Engineering and Physical Sciences Research Council through the Mathematics of Systems II Centre for Doctoral Training at the University of Warwick (grant number EP/S022244/1). We acknowledge the Scientific Computing Research Technology Platform at the University of Warwick for providing the compute cluster on which the experiments presented in this paper were run.

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

## A  Appendix

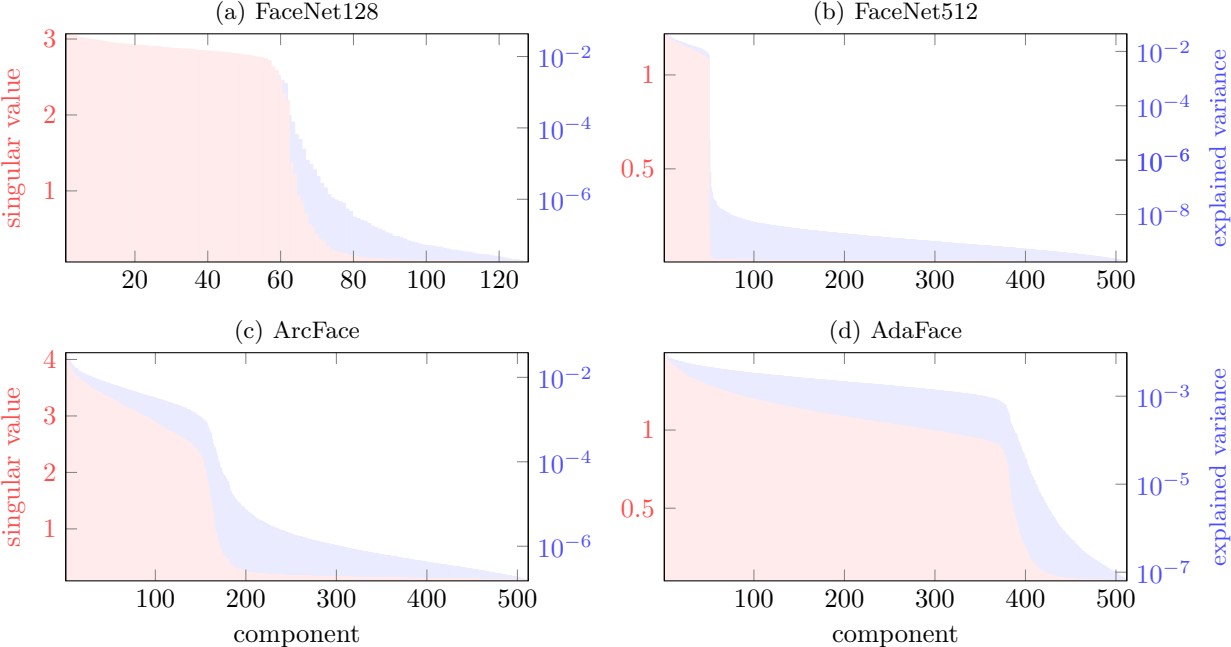

Figure 6: For each network, the singular values for the last linear weight matrix and the explained variance of the principal components of all images from the LFW dataset in feature space are overlaid for comparison.

Table 2: $(\varepsilon, \delta)$-dichotomy values for sets of feature vectors from various combinations of models and datasets after replacing each singular value of the last linear weight matrix with its mean.

|  | LFW | | | | CelebA | | | |
| --- | --- | --- | --- | --- | --- | --- | --- | --- |
| Model | $\varepsilon$ | $\delta$ | $i$ | $j$ | $\varepsilon$ | $\delta$ | $i$ | $j$ |
| FaceNet128 | 0.195 | 0.501 | 53 | 78 | 0.188 | 0.504 | 53 | 77 |
| FaceNet512 | 0.041 | 0.500 | 39 | 60 | 0.035 | 0.502 | 44 | 62 |
| ArcFace | 0.287 | 0.501 | 0 | 147 | 0.291 | 0.502 | 0 | 149 |
| AdaFace | 0.219 | 0.503 | 399 | 511 | 0.209 | 0.502 | 402 | 509 |

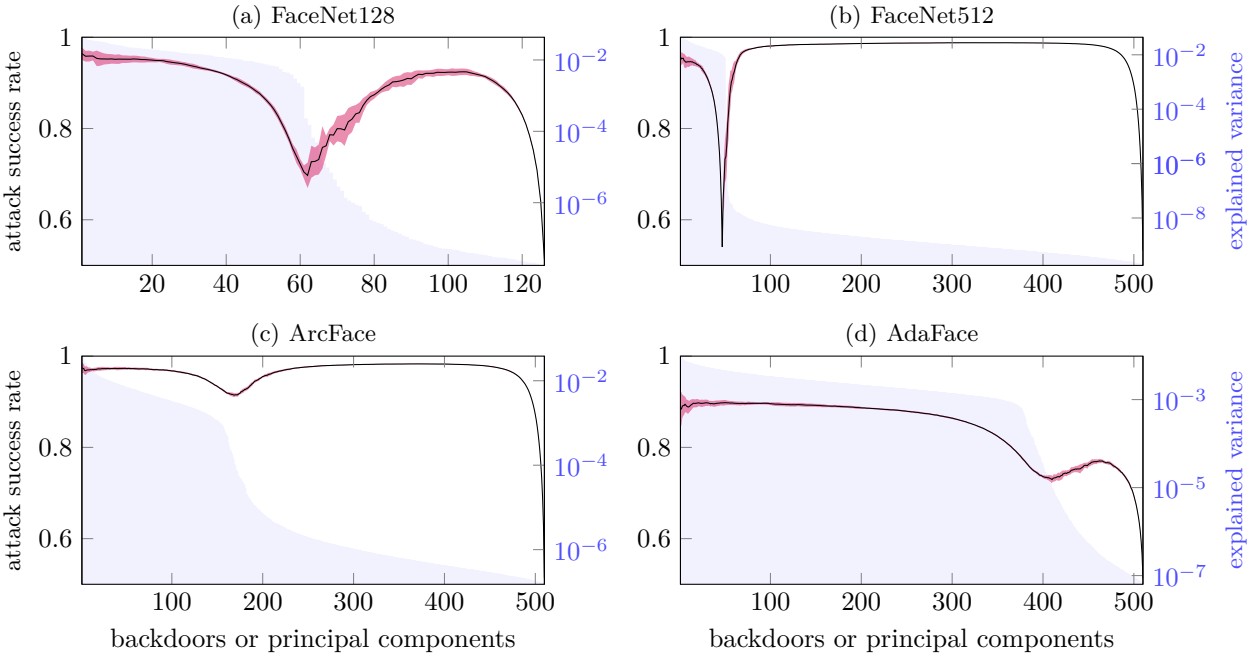

Figure 7: The black curves show the attack success rates after a number of Shattered Class backdoors are installed. The success rate of a backdoor is the proportion of pairs of inputs from its CelebA class or classes that are misclassified by the system. These rates are averaged over all the backdoors currently installed, over the ten standard benchmark 90%–10% splits with the LFW dataset. This is repeated ten times, with different random choices of backdoored CelebA classes, and the average is plotted. The red band shows the standard deviation. In addition, the shaded blue area shows the explained variance of the principal components of all images from the LFW dataset in feature space, on a logarithmic scale.

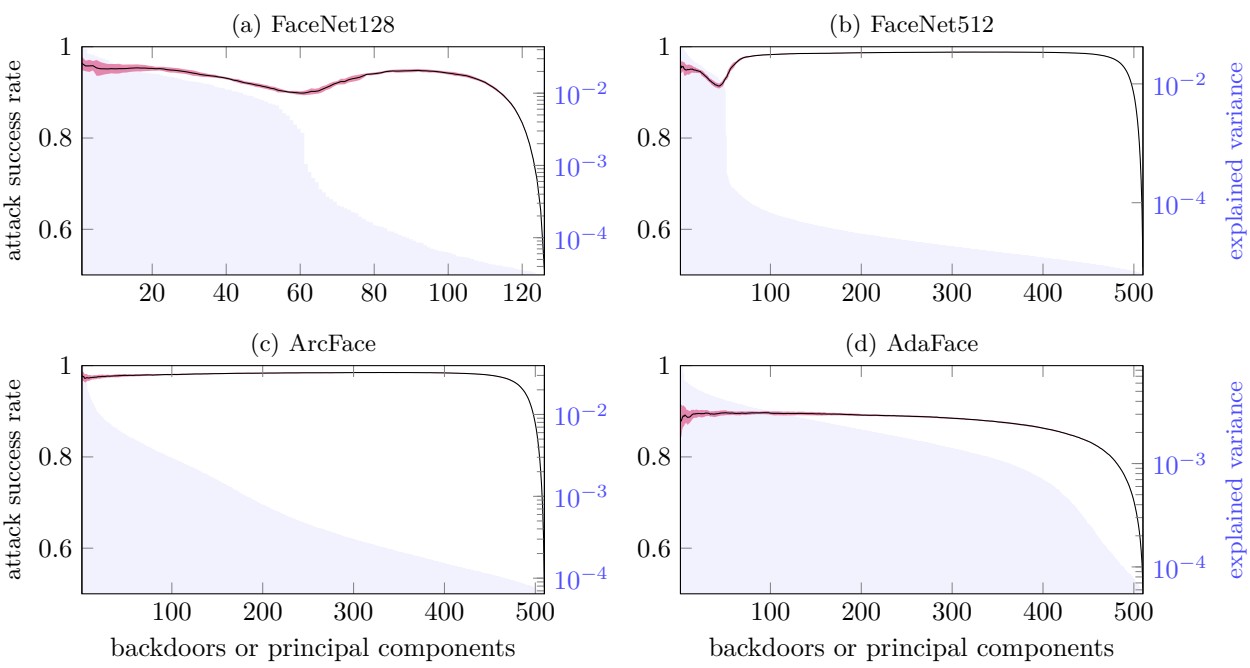

Figure 8: The plots show the results of the same experiments as in Figure 7, but after first performing SVD on the last network layer and replacing the singular values with their mean. As before, the black curve and the red band show the average benign accuracies, and their standard deviation, respectively. The shaded blue area shows the explained variance of the principal components of all images from the LFW dataset, computed after the weight surgery and on a logarithmic scale

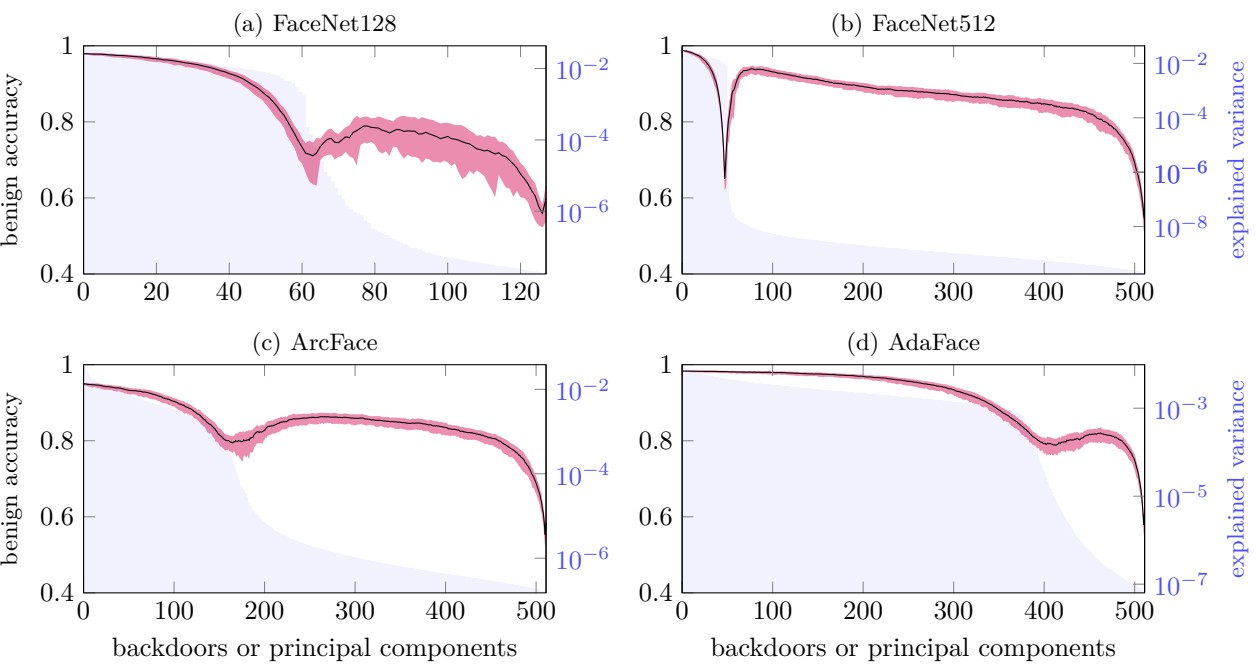

Figure 9: The black curves depict the benign accuracy of the networks as increasing numbers of concurrent Merged Class backdoors are installed, measured using the LFW dataset. Each backdoor applied is a Merged Class, with randomly picked individuals from the CelebA dataset (which have not been used in a previously installed backdoor) for which the backdoor is installed. This is repeated ten times and the average is plotted. The red band shows the standard deviation. In addition, the shaded blue area shows the explained variance of the principal components of all images from the LFW dataset in feature space, on a logarithmic scale.

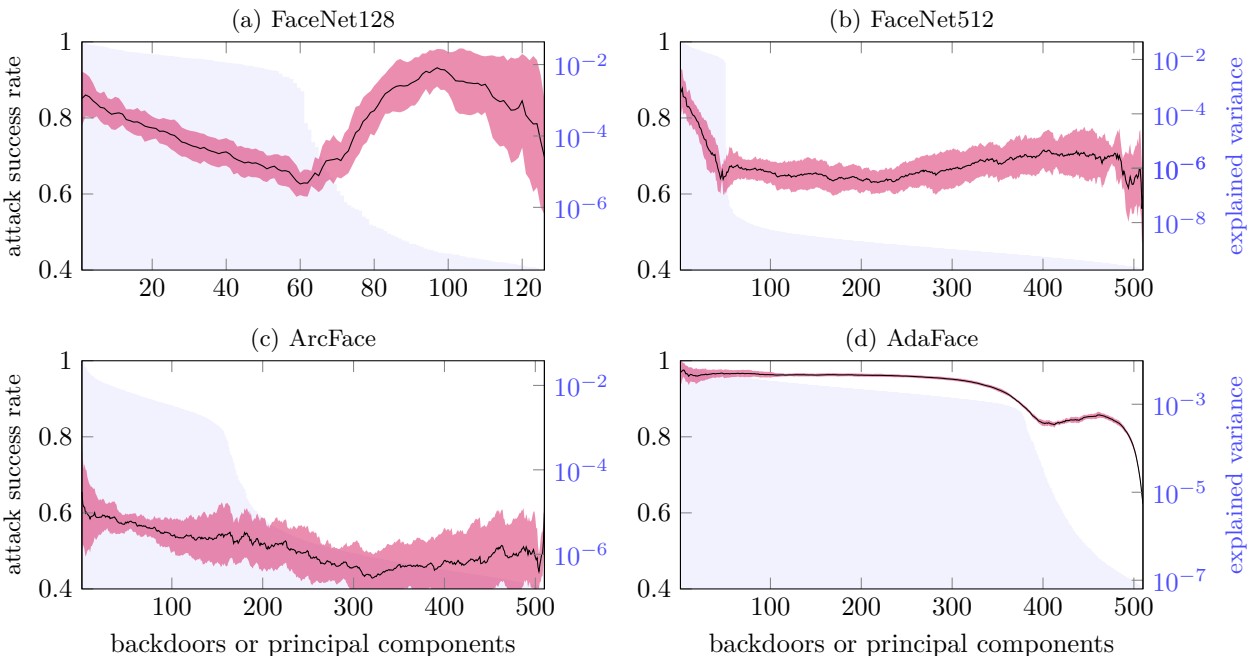

Figure 10: The black curves show the attack success rates after a number of Merged Class backdoors are installed. The success rate of a backdoor is the proportion of pairs of inputs from its CelebA class or classes that are misclassified by the system. These rates are averaged over all the backdoors currently installed, over the ten standard benchmark 90%–10% splits with the LFW dataset. This is repeated ten times, with different random choices of backdoored CelebA classes, and the average is plotted. The red band shows the standard deviation. In addition, the shaded blue area shows the explained variance of the principal components of all images from the LFW dataset in feature space, on a logarithmic scale.

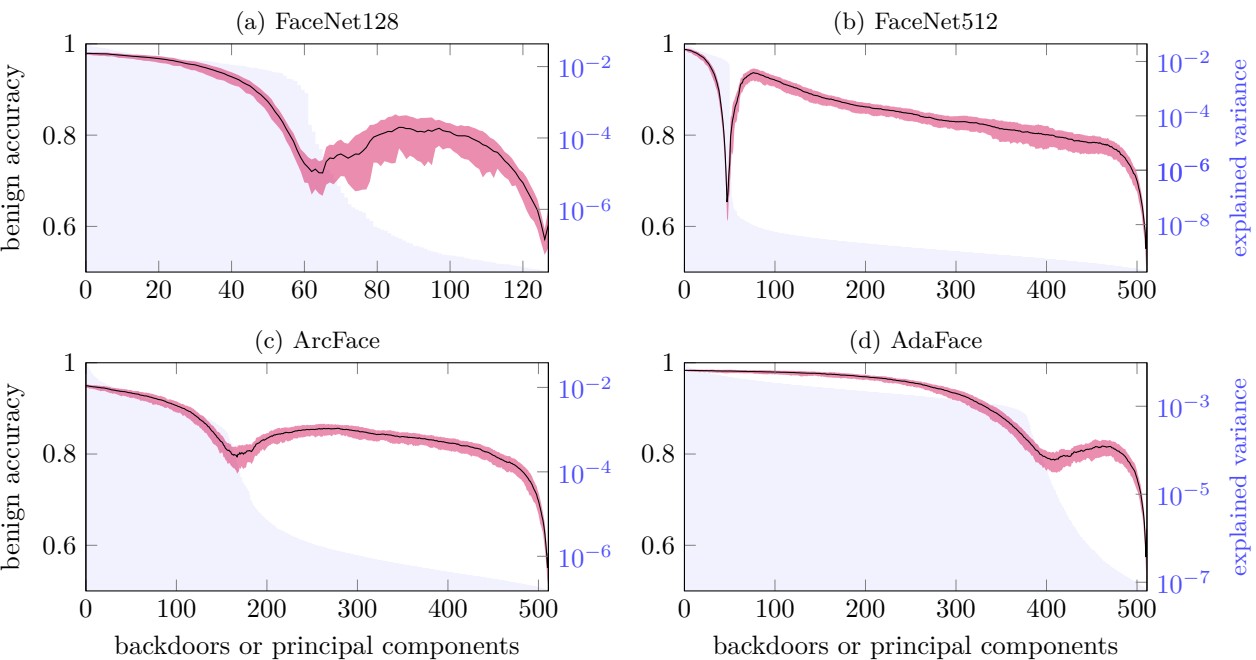

Figure 11: The black curves depict the benign accuracy of the networks as increasing numbers of concurrent backdoors are installed, measured using the LFW dataset. For each backdoor, a uniformly random attack mode (Shatter Class or Merge Class) is selected, and individuals from the CelebA dataset (not used in previous backdoors) are randomly chosen to install the backdoor. This process is repeated ten times, with the mean plotted and the red band indicating the standard deviation. The shaded blue area represents the explained variance of the principal components of all LFW dataset images in feature space, plotted on a logarithmic scale.

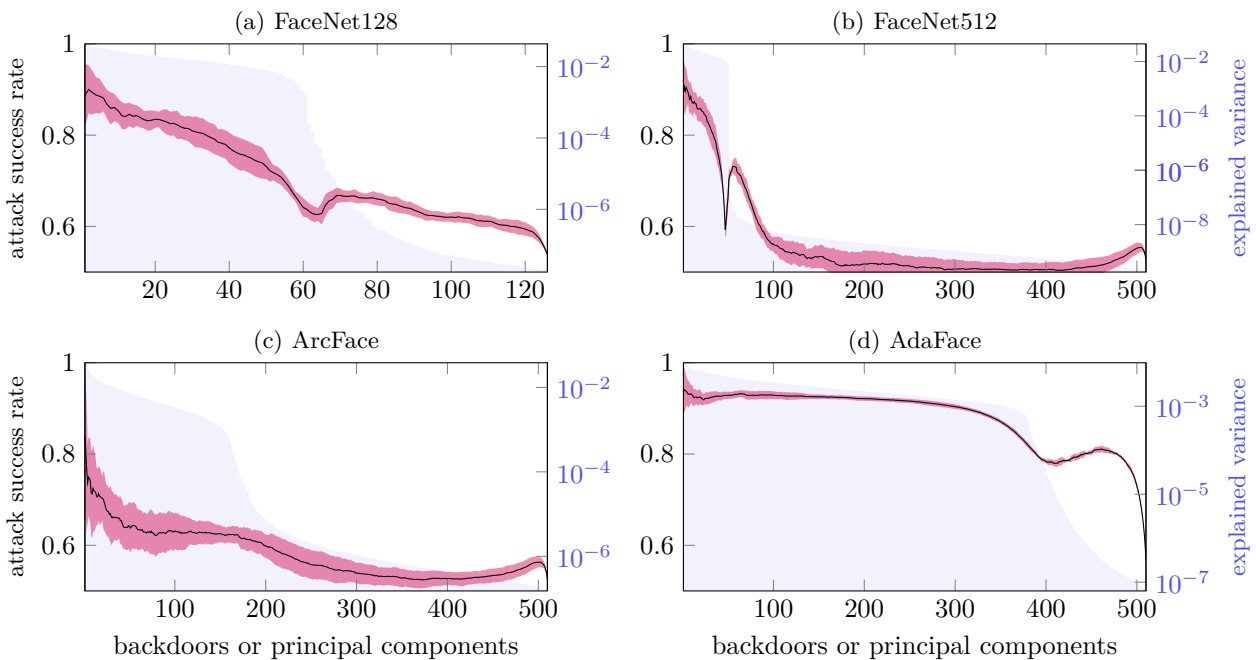

Figure 12: The black curves show the attack success rates after a number of backdoors are installed. The success rate of a backdoor is the proportion of pairs of inputs from its CelebA class or classes that are misclassified by the system. These rates are averaged over all the backdoors currently installed, over the ten standard benchmark 90%–10% splits with the LFW dataset. This is repeated ten times over random choices of the two backdoor types and the backdoored CelebA classes, and the average is plotted. The red band shows the standard deviation. In addition, the shaded blue area shows the explained variance of the principal components of all images from the LFW dataset in feature space, on a logarithmic scale.

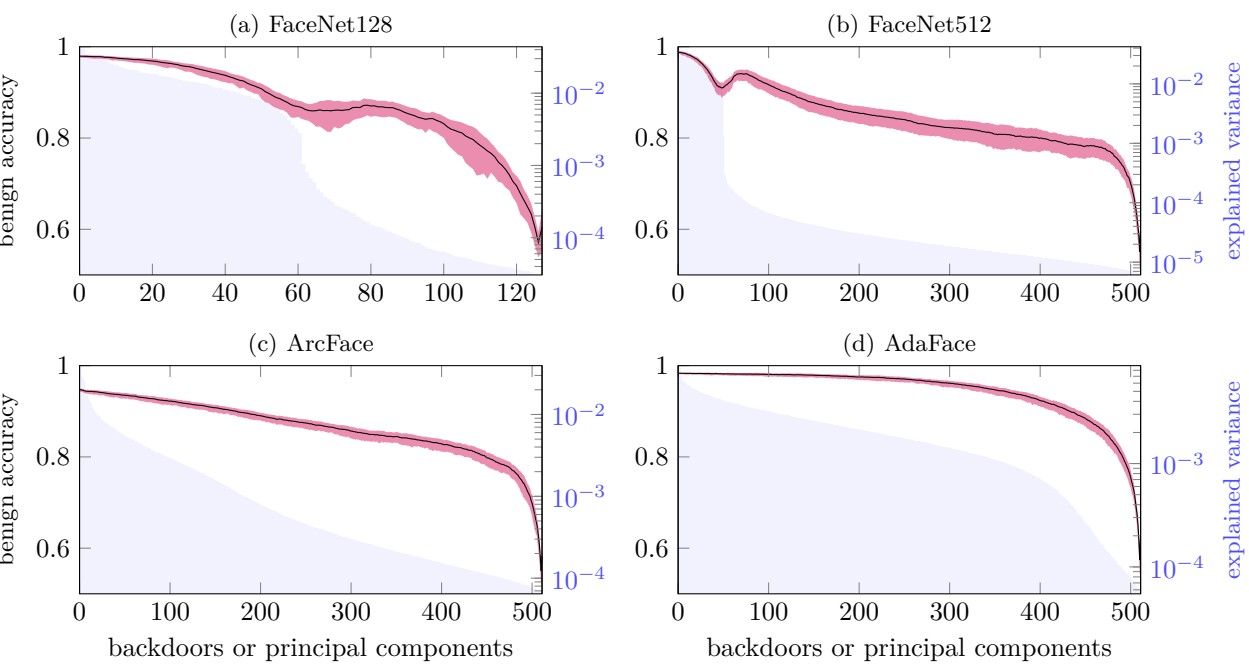

Figure 13: The plots show the results of the same experiments as in Figure 11, but after first performing SVD on the last network layer and replacing the singular values with their mean. As before, the black curve and the red band show the average benign accuracies, and their standard deviation, respectively. The shaded blue area shows the explained variance of the principal components of all images from the LFW dataset, computed after the weight surgery and on a logarithmic scale.

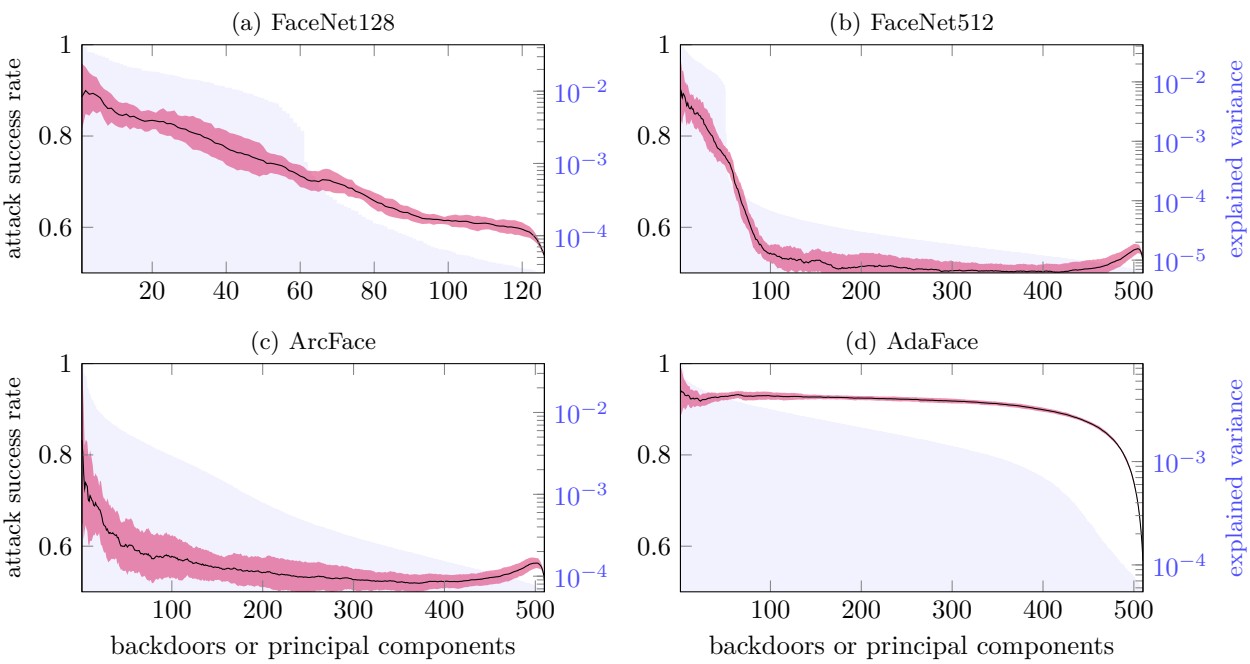

Figure 14: The plots show the results of the same experiments as in Figure 12, but after first performing SVD on the last network layer and replacing the singular values with their mean. As before, the black curve and the red band show the average benign accuracies, and their standard deviation, respectively. The shaded blue area shows the explained variance of the principal components of all images from the LFW dataset, computed after the weight surgery and on a logarithmic scale

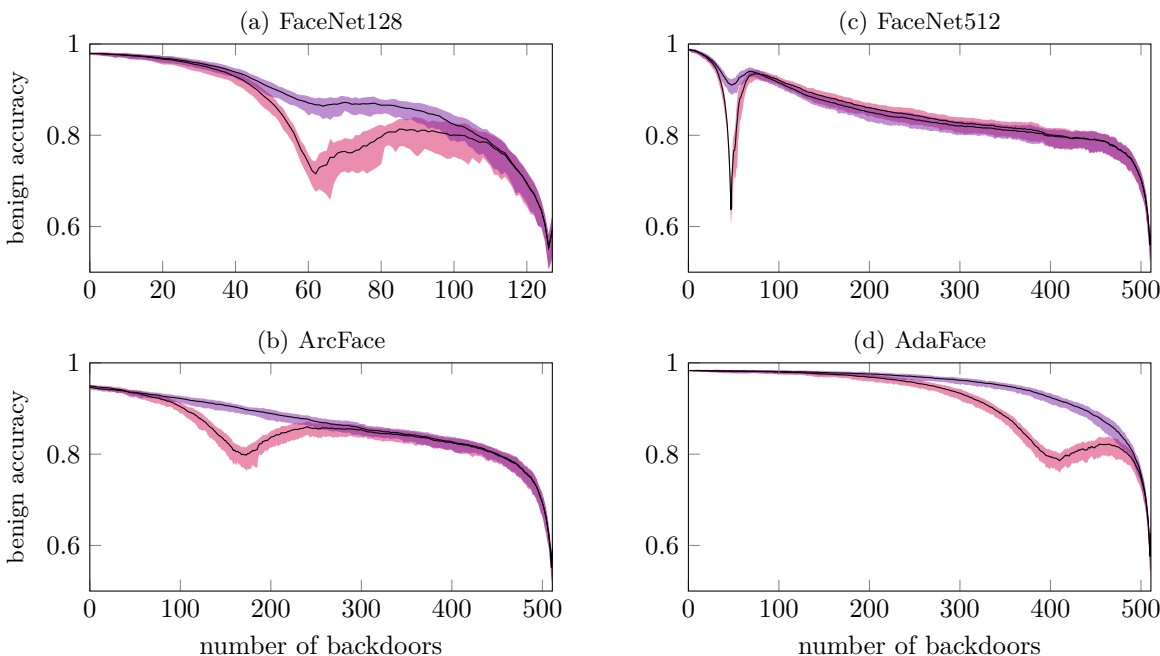

Figure 15: The benign accuracy from the original method, as shown in Figure 2, and from the revised method, as shown in Figure 4, are overlaid for easier comparison. For example, for FaceNet512 we see that after approximately 70 backdoors, the revised method performs better until around 300 backdoors at which point they are very similar again.

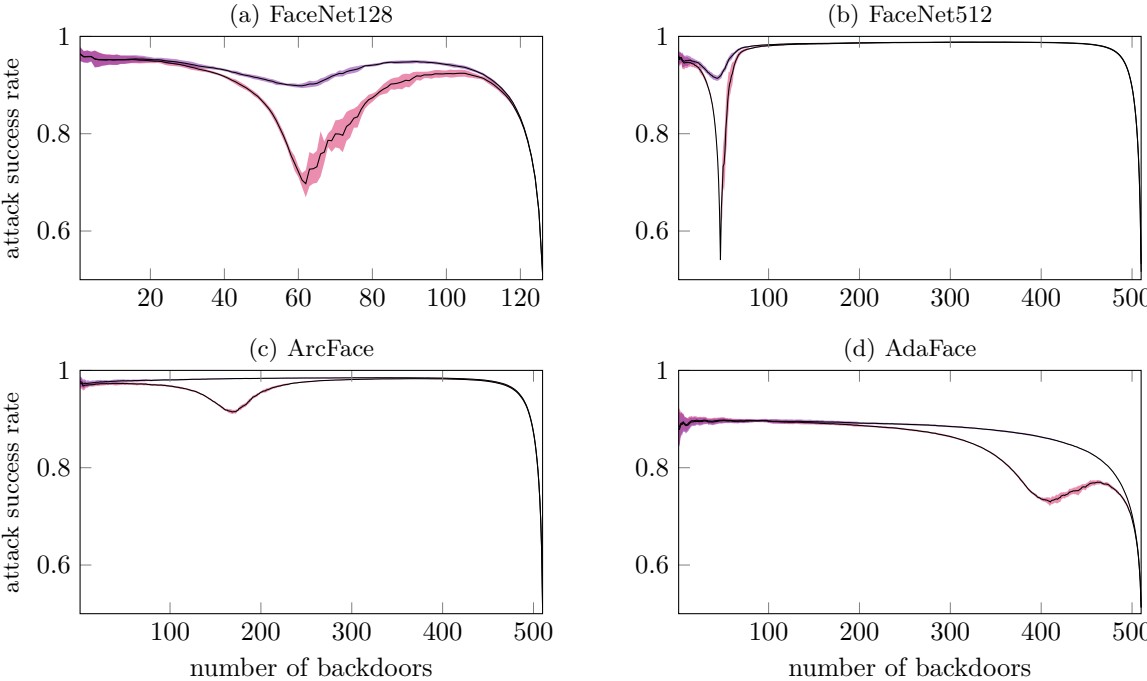

Figure 16: The attack success rate from the original method, as shown in Figure 7, and from the revised method, as shown in Figure 8, are overlaid for easier comparison.

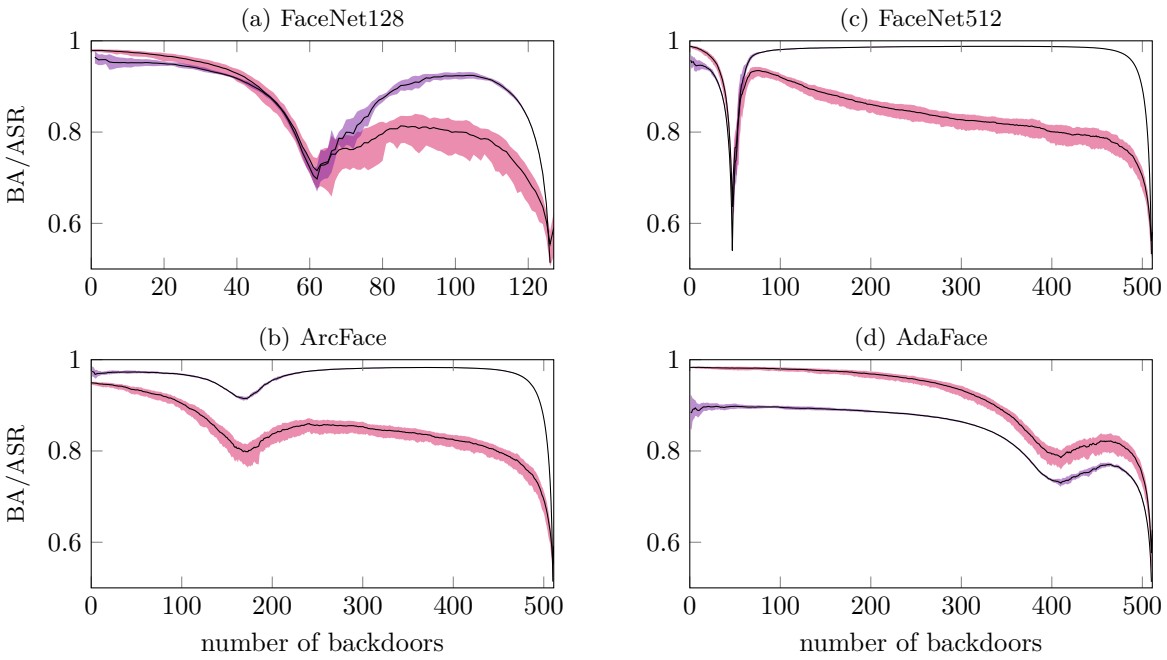

Figure 17: The benign accuracy (red) from the original method, as shown in Figure 2, and the attack success rate (purple), as shown in Figure 7, are overlaid for easier comparison.

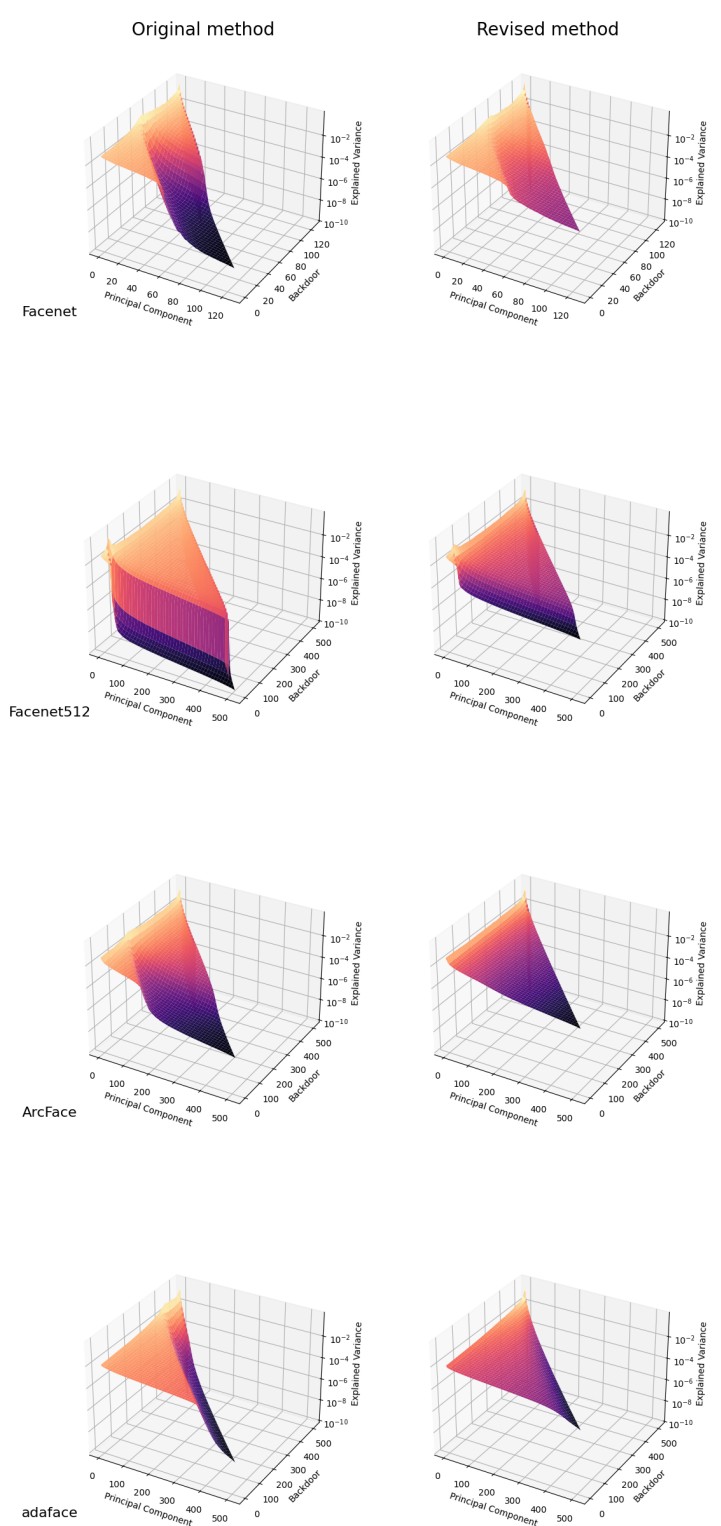

Figure 18: These surface plots show the explained variances of LFW dataset feature vectors as increasing numbers of concurrent Shattered Class backdoors are installed, both for the original method and our revised method. Other experimental details remain the same as in Figure 2.

