# OpenReview forum: "Variance Dichotomy in Feature Spaces of Facial Recognition Systems is a Weak Defense against Simple Weight Manipulation Attacks"
_TMLR — Accepted by TMLR_

### Review · Reviewer_SRe9 · 2025-03-20

**Summary Of Contributions:**

This paper analyzes the shortcomings of backdoor attacks on simple weight manipulations in only the last hidden layer. Through empirical experiments, the authors found that they exhibit a variance dichotomy in their feature spaces, which causes the benign accuracy of the attacked system to decrease fast as the number of sequentially installed backdoors increases. Finally, the authors proposed an improved backdoor attack.

**Audience:**

Yes

**Broader Impact Concerns:**

None.

**Claims And Evidence:**

No

**Requested Changes:**

See Strengths And Weaknesses.

**Strengths And Weaknesses:**

Strengths:
1. The discovery of ​variance dichotomy and its link to benign accuracy double descent is a significant contribution. This phenomenon explains why prior attacks (e.g., Zehavi et al.) face diminishing returns as backdoors accumulate, advancing understanding of intrinsic defense mechanisms in facial recognition systems.
2. The work highlights critical vulnerabilities in widely used facial recognition systems, urging the community to address weaknesses in feature space design. The discussion of "intrinsic dimension" and simplicity bias bridges gaps between security and representation learning.

Weaknesses:
1. The theoretical explanation of the double descent phenomenon still relies on simplified models (such as random vector projection) and lacks rigorous mathematical proof.

2. The causes of variance dichotomy (such as the influence of training objectives and loss functions) have not been explored in depth, which may affect the universality of the conclusions.

3. Although this paper discovered the phenomenon of variance dichotomy, its intuitive explanation was not given.

4. The expression of this paper needs further improvement. For example, the background and significance of the work are not very clear at present, and the correction of only one attack limits the influence of the paper. The experiments and methods are always expressed together, which is not conducive to everyone's understanding. In addition, the contribution should be simply summarized instead of lengthy output.

---

> ### Author Response · Authors · 2025-04-11
> **Thank you very much for your review.**
>
> Many thanks for the thoughtful comments and questions, which we think have resulted in an improved revised manuscript. Please find some responses below.
>
> 1\) The current paper is not primarily a theoretical paper. However, we would like to see a mathematical proof (possibly in the simplified setting we describe) and mention this as an exciting direction for future work.
>
> 2+3) The revised paper now gives a lot more intuition about the dichotomy phenomenon and its connection to low-dimensionality. We thank the reviewer for this comment as we believe this has greatly improved our presentation and hope the reviewer also finds the new text useful.
>
> Beyond that, we agree that it would be very interesting to study why the variance dichotomy emerges and why the non-dominant feature dimensions still contain meaningful information. We do cite Ansuini et al. and Morwani et al. as an example of two papers that study phenomena that may be related. Giving a full explanation is an exciting future research direction. Given that we can empirically see that quite a few facial recognition systems have this property, we believe that the phenomenon is not universal, but very common.
>
> 4\) Thank you again. We made significant changes to the introduction. We also reorganized the paper to introduce the dichotomy phenomenon first, because we think this is of wider interest. We have also separated results and our general setup from more technical descriptions about precise details of the experiment. The latter can now be found in Section 3.1.

---

### Review · Reviewer_Urxe · 2025-03-21

**Summary Of Contributions:**

This is an empirical paper providing a more in depth analysis of the adversarial attack introduced by Zehavi et al. (2024), which involves manipulating the weights of the last layer of a deep neural network, to affect the capabilities of a facial recognition system on specific targets. The paper mainly explores how many backdoors can be inserted in the model (number of targets) without eccessively damaging the performance of the model on independent samples, framed as benign accuracy, and computed on a separate dataset. The authors identify a maybe surprising double descent behaviour of the benign accuracy with respect to both the benign accuracy and the success of the attack rate (the proportion in which a pair of the target dataset is misclassified).

**Audience:**

Yes

**Claims And Evidence:**

Yes

**Requested Changes:**

My major requested changes regard Weakness number 2. I would appreciate if the authors introduce more definitions, maybe formalizing the exact quantities in the plots via equations and proper notation. The phenomenon investigated in this paper is in my opinion, very interesting, but requires a solid setting definition, also to enable later theoretical investigation by future work.

**Strengths And Weaknesses:**

**Strenghts**

I believe this paper to present a very interesting phenomenology. This double descent phenomenon as a function of number of backdoors is, in my opinion, very interesting, and quite unexpected. The authors present their intuition, which connects this non-regular descent to a sort of spectral gap in the covariance of the features, and their proposed method to remove this "bump" seems to be working, supporting their hypothesis.

**Weaknesses**

I believe this paper to have quite a few weaknesses, listed below:

1 - At the moment, the narrative has a relative restricted scope. In other words, the open question this paper aim to address is the one at the end of page 2, which is very specific and very focused on an open istance left behind by Zehavi et al. (2024). From the perspective of a reader without strong familiarity with the area, this research goal is narrow and fairly not motivated. I recommend the authors to modify the introduction to make the narrative more engaging, for example describing the setting and the experiment from the very beginning, discussing the surprising phenomenology of Figure 1, and leaving the study and understanding of this problem as the purpose of the work. This is not a major request, I leave to the authors the ultimate choice regarding this.

2 - The setting of the experiments is often not very clear. First, the _explained variances_ firstly introduced in Section 1.1 are not preoperly defined or linked to any definition. Even when better introduced in Section 3, the expanation is not very smooth: why do the authors mention PCA (that should generally more refer to a dimensionality reduction technique)? Are the authors computing the empirical covariance of the features of the penultimate layer on the target dataset? How large is this dataset and how large is the space of these features? I presume the authors report the renormalized eigenvalue of such empirical covariance.

Regarding the same point, at point 4 in Section 1.1 (later remarked in the last paragraph of Section 2) the Authors mention that they take random choices of the attack. What does this mean? This, to me, seems like a very confusing variable in trying to understand what are the causes of the double descent phenomenon. Why did the Authors follow this choice, rather than focusing on one single attack, or carrying the study separately? Furthermore, computing the accuracy of the two attacks should be qualitatively different (the first one is in-class, while the second one regards two target classes). Thus, it is also not obvious to me how the attack success rate is exactly computed in Figure 2, for example. Also, how is the cosine similarity threshold computed? To optimize precision or recall? On which attack type?

---

> ### Author Response · Authors · 2025-04-11
> **Thank you very much for your review.**
>
> We thank the reviewer for the review and, in particular, for the two main detailed suggestions. We made changes to address these as follows and think this has improved the presentation of the paper:
>
> **“the narrative has a relative restricted scope”**
>
> In response to this concern, we have significantly rewritten our introduction and changed the ordering of sections within the paper. We now introduce and analyze the concept of variance dichotomy in facial recognition systems first. We then discuss the impact of this on the attack scheme by Zehavi et al. (2024), and introduce and explain the double descent phenomenon we uncover. Finally, we explain how this analysis can be used to increase the performance of the attack scheme. We believe these changes have improved the clarity and impact of our paper and appreciate the reviewer’s helpful suggestion in this regard.
>
> **“The setting of the experiments is often not very clear”**
>
> Thank you for making a number of suggestions for improvement. We have made several changes which indeed result in a better and more understandable presentation.
>
> First, we have explicitly introduced and defined both explained variances and principal components. We believe this helps to clarify that while PCA is often associated with dimensionality reduction, in our case, we leverage its decomposition to analyze the variance structure of feature space.
>
> Second, in Section 2, we have refined the explanation of our experimental procedure. We explicitly describe that we compute the empirical covariance of the feature vectors corresponding to the LFW dataset. This dataset contains 13000 images and the dimension of feature space is 512 for all networks apart from Facenet128, for which it is 128.
>
> Indeed, we do report the renormalized eigenvalues of the empirical covariance matrix, as the reviewer correctly inferred. This is made explicit in our revised explanation.
>
> In the revised version of the paper, we have moved the figures focusing solely on the shattered class backdoor into the main text. Regarding "random choices," we clarify that this refers to selecting between shattered class and merged class backdoors uniformly at random for each installation. We believe these experiments are valuable to the reader, as they reflect a realistic attack scenario. However, to improve clarity around explaining the double descent phenomenon, we now present the figures using a mix of backdoor types in the appendix and focus on only the shattered class backdoor in the main text.
>
> Indeed, the attack success rates are calculated differently for each backdoor type. For a SC backdoor, we are interested in the success rate in terms of two images of the target individual to be declared a mismatch. For the MC backdoor, we are interested in the success rate in terms of two images of the respective target individuals to be declared a match.  In order to calculate the overall attack success rate we calculate the attack success rate for each backdoor individually and calculate the mean. The value we are presenting in Figure 2 therefore reflects how often a backdoor installed by an attacker successfully achieves its intended effect.
>
> The cosine similarity threshold is computed to maximize the accuracy on a subset of the LFW dataset. This dataset contains an even split of pairs of images that are matched (i.e. of the same person) and mismatched. The cosine similarity threshold we pick is the one that best separates matched pairs from mismatched pairs. We are therefore indirectly balancing precision and recall.

---

> > ### Comment · Reviewer_Urxe · 2025-04-27
> >
> > I would like to thank the Authors for their revision, and I apologize for the late feedback.
> >
> > I agree with the Authors that this (consistent) revision strongly improved the clarity of their work. I largely appreciated the very extended Sections 1 and 2, and the addition of Figure 1, which, in my opinion, provides the right intuition for the reader to be ready to digest the later results.
> >
> > To be precise, I also found Definition 1 to be useful to completely follow the Authors arguments. Maybe one natural question now is why is it necessary / important to renormalize every entry of $X$ to obtain $\hat X$, why not simply computing the true covariance of $X$? Also, is Definition 2 inspired by some prior work? It is a reasonable definition, but I wonder if the Authors built it from scratch or if it is motivated by prior definitions to characteriza spectral gaps in high dimensional matrices...
> >
> > To conclude, I appreciate the decision of the Authors to focus on the shatter class attack in the main body. I do believe this choice to reduce the number of moving parts in the experiment and to provide a simpler setting that is rich enough to showcase the intended phenomenology. I will edit my evaluation on the "claims and evidence" field.

---

> > > ### Author Response · Authors · 2025-06-19
> > >
> > > We agree with the reviewer’s observation regarding the standardization step in Definition 1. We have removed the normalization by standard deviation across components when computing the covariance matrix. Importantly, this had a negligible effect on the resulting explained variances and all key conclusions are unaffected by this change.
> > >
> > > Definition 2 is not inspired by any prior work. To the best of our knowledge, there are no existing definitions in the literature that specifically characterize spectral gaps in the way we do here.

---

### Review · Reviewer_9v8r · 2025-03-24

**Summary Of Contributions:**

This paper analyzes an existing attack that manipulates the weight of the last layer in a victim facial verification network and observes a phenomenon called variance dichotomy. When the number of injected backdoors increases, the benign utility first decreases and then increases before finally decreasing. Based on this observation, the authors propose a way to transform the weight of the last layer before injecting the backdoors to mitigate this dichotomy phenomenon which can be used as a potential defense against that existing attack. Experiments are conducted on four models with LFW and CelebA.

**Audience:**

Yes

**Claims And Evidence:**

No

**Requested Changes:**

1. It's suggested that the goal of this paper be clearly stated and that generalizable observations and results be provided.

2. From my understanding, the explained variance is dependent on the last layer's weight and the inputs according to the description in Section 3. That is, it changes after injecting each backdoor. Then, how can one draw the shade for the variance in Figure 1-2? Because when the backdoors increase along the x-axis, the variance matrix should also change.

3. If the x and y axes are the same between Figures 1 and 2, the two figures may be merged for easier comparison between the trends of benign utility and backdoor ASR.

4. It's unclear why the authors randomly chose two different backdoors. If one backdoor is used, will the same phenomenon be observed?

5. According to Definition 1, $\epsilon$ and $\delta$ are not unique for each network and dataset. How are they decided, and why does the paper choose $\delta=0.5$? Why are the values of $\epsilon$ for Facenet512 in Table 1 0?

6. In Section 4, is SVD needed for each backdoor? For example, injecting the first backdoor requires SVD on the benign weight. Does injecting the second backdoor require SVD on the already modified weight?

7. Could you further explain why a synthetic dataset is needed in Section 5? It may not correspond to any real-face dataset. What extra evidence does it provide to support the paper?

4. It would be better to mention some defenses in the related work. Also, since the authors mentioned physical adversarial attacks in the related work, it's suggested to add https://ieeexplore.ieee.org/document/10179360

**Strengths And Weaknesses:**

Strengths:
1. Overall, it's easy to follow.
2. Evaluated four models.

Weaknesses:
1. The goal of this paper is unclear. Is it a defense or an attack? From the title, it seems a defense. However, from the main text, it seems an attack that improved the existing attack. Or is it just analyzing one existing attack? What is the takeaway to the audience?
2. Because the paper only talks about one existing attack, the analysis and results may not be generalized to other attacks.

---

> ### Author Response · Authors · 2025-04-11
> **Thank you very much for your review.**
>
> Many thanks for your careful reading and, in particular, the many concrete suggestions and questions. We revised the paper accordingly and think this has helped in improving the presentation.
>
> **"The goal of this paper is unclear. Is it a defense or an attack?”**
>
> We revised the introduction and hope this is now more clear. The message is twofold: (1) We highlight a particular property that a number of facial recognition systems have. Namely, the feature vectors approximately lie in a much lower dimensional linear subspace of the entire feature space. Interestingly, the systems we consider maintain a high accuracy when we project the feature vectors onto the orthogonal subspace, suggesting that this subspace still contains useful semantic information. These facial recognition systems were not intentionally designed this way. It is merely an anomaly, if you will, of the training process. We believe this observation to be of independent interest. (2) We then observe that this property is what stops the attack by Zehavi et al. (2024) from realizing its full potential. Given this, we then demonstrate an improved attack.
>
> In summary, we do not propose a defense, but only identify which property of facial recognition systems limits the success of the attack by Zehavi et al. We then propose an improved attack to overcome that limitation.
>
> **“the analysis and results may not be generalized to other attacks”**
>
> We hope that the variance dichotomy property we observe in a number of different systems is of independent interest beyond this particular attack.
>
> ### Response to requested changes:
>
> 1. We revised the introduction, reorganized the paper and now also provide more explanations and definitions.
>
>
> 2. What we draw are the values prior to any backdoor being installed. We edited the caption to make this more clear. We have now added Figure 18 to the appendix to show how the explained variances change as we install shattered class backdoors. The figure shows how the number of large explained variances reduce by roughly one with each backdoor, whilst we still maintain the same number of small explained variances. After a number of backdoors roughly equal to the number of "large" explained variances we no longer have a dichotomy. This is the point in our experiments at which we get a large increase in benign accuracy.
>
>
> 3. We have now added the merged version of these figures to the appendix (Figure 17).
>
>
> 4. Thank you. Yes, the same phenomenon is observed using only one backdoor type. In the appendix we previously provided plots for only one or the other backdoor. We now moved the plot of the experiments only involving Shatter Class backdoors to the main body of the paper (and moved the mixed backdoor plot into the appendix) because we think that you are entirely correct in pointing out that mixing backdoors creates additional complications and risks obfuscating the main point we are making. We keep the mixed backdoor plots in the appendix, because we still think that it is a realistic scenario that one or more attackers may want to install different types of backdoors in the same system.
>
> 5. Thank you. We added additional clarifications to the text. We pick $\delta=0.5$, because this value guarantees that there can only be a single drop of this magnitude and therefore, we have a true dichotomy. We then find the smallest range of the explained variances over which a drop of at least this magnitude occurs, and the fraction of the explained variances that this range covers is our calculated $\varepsilon$. $\delta$ sometimes varies between networks due to us having a discrete set of explained variances. As an example, Facenet128 has a drop of magnitude 0.5 between the explained variances 60 and 73 when using the LFW dataset. The shaded blue region in Figure 1 shows this visually, between the indices 60 and 73 we have a large drop in explained variance. The values of $\varepsilon$ for FaceNet512 were 0, because we rounded all values to two decimal places, the drop is very sharp and occurs over only two principle components, and 2/512 is approximately 0.0039. In the revised version of the paper, we have rounded values to three decimal places for better clarity.

---

> > ### Author Response · Authors · 2025-04-11
> > **Part 2 of our response.**
> >
> > 6. Thank you. We have clarified this in the revised version. Manipulating the singular values is done only once at the start. After performing this operation, the feature vectors belong to a subspace with dimension approximately equal to the full dimension of feature space. This alone is enough to eliminate the dips seen in benign accuracy when installing multiple backdoors. Figure 18 is useful to see why it is not necessary to reapply SVD for each backdoor. After we have applied SVD we have largely reduced the dichotomy in the explained variance, this can be seen by looking at the explained variances after the first backdoor in the revised method. As we install more backdoors we do not get large changes to individual explained variances. Since we do not have the reappearance of a large dichotomy in the variances it is not required to perform the SVD modification again.
> >
> >
> > 7. We believe it helps to illustrate the main cause behind the double descent phenomenon observed, i.e., why the benign accuracy first decreases as successive backdoors are installed, but then, surprisingly (at least to us), increases again just to decrease for a second time. Specifically, that this is not primarily caused by any very particular property of the dataset or any complex part of the network, but can mostly be explained by the dichotomy property of singular values in the last network layer alone.
> >
> >
> > 8. Thank you for suggesting this. We have now included a section on adversarial defenses in the related work section. We have also added the suggested citation.

---

> > > ### Comment · Reviewer_9v8r · 2025-04-30
> > >
> > > Thank the authors for the response and revision. Most of my concerns have been addressed.

---

### Decision · Action_Editor_N4iV · 2025-05-19

**Recommendation:** Accept as is

**Comment:**

The AE highly encourages the authors to address the following two remaining concerns:

1. Clarification of Definition 2, as requested by the reviewer: https://openreview.net/forum?id=Q1Cf07flwD&noteId=dEEFCtTHIk
2. Addition of a discussion section at the end of the conclusion, outlining potential directions for future work, specifically, how one might further investigate the phenomenon through extensive evaluations and explore possible theoretical connections.

Additionally, the AE has a minor suggestion regarding the title:

1. Weak Defense from -> Weak Defense against

**Audience:**

The main finding---a connection between the variance dichotomy and the double descent curve in benign performance---is intellectually stimulating and well-suited for TMLR's audience, with the potential to inspire future research in the field.

**Claims And Evidence:**

This is an empirical paper that explores the connection between the variance dichotomy and the double descent curve in benign performance, particularly in the context of backdooring facial recognition models. Some reviewers raised concerns regarding the lack of theoretical justification and requested a more comprehensive evaluation. However, the AE read the work of Zehavi et al. and found the breadth of experiments in the current paper to be sufficient. I also agree with the authors’ response that this is not intended to be a theoretical paper.